# Clinical Efficacy of 1% CHX Gluconate Gel and 0.12% CHX Solution: A Randomized Controlled Trial

**DOI:** 10.3390/ijerph19159358

**Published:** 2022-07-30

**Authors:** Su-Young Lee, Eun-Joo Nam

**Affiliations:** 1Department of Dental Hygiene, Namseoul University, Cheonan-si 31020, Korea; 2Department of Dental Hygiene, Graduate School, Namseoul University, Cheonan-si 31020, Korea; dt2@nsu.ac.kr

**Keywords:** BANA test, chlorhexidine, gingival inflammation, oral health

## Abstract

Chlorhexidine (CHX) is an effective antibacterial agent and is used in dental treatment in several formulations. The aim of this study was to compare the effectiveness of CHX solution and CHX gel on dental plaque inhibition and gingivitis relief by a randomized clinical trial. Thirty-eight participants were randomly divided into two groups: control group (0.12% CHX solution) and test group (1% CHX gel). Participants were provided with CHX products and were instructed to use each product in the morning and evening for 1 week. Clinical results were evaluated by analyzing the collected data of Turesky et al. the modified Quigley-Hein Plaque Index (TQHPI), gingival index (GI) and the BANA test. Measurements were conducted 4 weeks and 8 weeks after using chlorhexidine products. The results were analyzed using repeated measured ANOVA and paired *t*-test. TQHPI and GI were significantly different after treatments in both groups (*p* < 0.001). The GI decreased more in the test group compared to the control group 4 weeks and 8 weeks later. In both groups, the BANA score also significantly decreased (*p* < 0.001) after 8 weeks, though the BANA score decreased relatively more in the CHX gel group than the CHX solution group. These results suggest that 1% CHX gel is more effective in reducing gingivitis and bacteria of periodontal disease than the 0.12% CHX solution. Therefore, the 1% CHX gel is expected to be actively used for non-surgical treatment of periodontal disease patients.

## 1. Introduction

Periodontal disease is typically a multifactorial disease that can result from a variety of causes, the main one being microorganisms forming biofilms in the oral cavity. It has been reported that *Actinobacillus actinomycetemcomitans*, *Porphyromonas gingivalis*, *Prevotella intermedia*, *Bacteroides forsythus Tannerella forsythia* and *Treponema denticola* mainly detected in subgingival dental plague are the main causative organisms for periodontal disease [1]. Treatments for periodontal disease usually involve removal of calculus on the surface of teeth and dental plaque by scaling and root planing. However, mechanical methods cannot completely remove the causative bacteria of the periodontal disease [2]. It has been reported that the causative bacteria of periodontal disease are reattached to 30% of the treated root surface and are covered by the remaining calculus after subgingival calculus removal [3]. Therefore, chemical treatment and physical treatment are combined for effective periodontal treatment [4].

Chlorhexidine (CHX) is considered the most effective antimicrobial to chemically control plaque [5]. It is a chemical synthesis disinfectant with broad-spectrum antiseptic action, active against Gram-positive and Gram-negative bacteria and also against fungi [6,7]. CHX inhibits the formation of dental plaque and relieves gingivitis by degeneration of the bacterial adhesion mechanism, denaturation of the bacterial cell wall, and dissolving cells [8]. CHX treatment also causes profound shifts in microbiota composition and metabolic activity. Total organic acid production in biofilms greatly decreased and mainly produced lactate after CHX-treat [9]. In a previous study [10], the effect of CHX was demonstrated by reducing periodontal pockets and gingival bleeding as a result of using CHX in the treatment of root planing in patients with chronic periodontal disease. It was reported in a systematic review that both 0.12% CHX and 0.2% CHX were effective in helping reduce gingival index and dental plaque [11].

CHX can be used in different formulations such as a solution, spray, gels, creams or toothpaste. Previous research has reported that it can be more effective for the control of dental plaque when applied in a solution than a spray [12]. However, a CHX solution may cause staining of soft tissues and dental surfaces in the oral cavity when used for a long period of time. To reduce these side effects, CHX has been developed and used in the form of gels and varnishes. A study using CHX gel and toothpaste reported that tooth stain, a side effect of CHX, was not found [13]. Recently, the preventive effect of CHX gel on the dry socket was investigated [14], and was also used for anti-infective treatment of peri-implant mucositis [15]. As such, CHX gel is widely used as an effective antimicrobial method for various oral treatments. However, the frequency of use of CHX solution is still much higher in dental clinics in Korea. In order to increase the utilization of CHX gel, which has no side effects and is effective, it is necessary to actively study and report the clinical effect of CHX gel. The null hypothesis of this study is that there is no difference in the clinical effects of CHX gel and CHX solution.

Therefore, the purpose of this study was to compare the clinical effect of the 1% CHX gel and 0.12% CHX solution for relieving gingivitis, inhibition of dental plaque formation, and antibacterial effects in patients with gingivitis.

## 2. Materials and Methods

### 2.1. Study Design and Subjects

This study was conducted as a randomized controlled clinical trial, and participants were selected adults who voluntarily applied through the clinical trial notice. The sample size was estimated as 38 using the G*Power program, based on the following conditions: two measurements through repeated measures ANOVA, effect size of 0.25, significance level of 0.05, and power of 0.85. Fifty-one participants were screened for this study, and two people were excluded according to the inclusion and exclusion criteria. The selected 49 subjects were randomly divided into two groups (control group: 25, test group: 24). During the study period, 11 patients were excluded for reasons of giving up or not visiting the clinic center (Figure 1). The final 38 subjects of this study were 25 women and 13 men, and the average age was 46.3 years.

Inclusion Criteria:adults of both sexes over 20 years,medically healthy subjects,at least 20 teeth remaining,understanding of the content and purpose of the study.

Exclusion Criteria:poor oral hygiene,an allergic reaction to chlorhexidine,≥5 mm periodontal pocket depth (PPD).

### 2.2. Clinical Protocol

This study was a randomly classifying the subjects into two groups. They were given CHX products to be used for 7 days: 0.12% gluconate CHX solution (Hexamedine, Bukwang Pharm.co, Ltd., Seoul, Korea) for control group and 1% CHX gluconate gel (CHoral GEL, All in one bio, Seoul, Korea) for test group.

The CHX solution group was instructed to use 15 mL and gargle for 1 min twice a day, whereas in the CHX gel group, according to the manufacturer’s instructions, an appropriate amount of the gel was applied to the inflamed gingival area using a finger and was spit out after 1 min. This was also performed twice a day. In addition, subjects were asked not to use oral health supplements or other antibacterial products and were instructed to apply the same brushing method with the same toothpaste. To evaluate the oral status of the subjects, plaque and gingival examination were performed at the baseline, 4 weeks, and 8 weeks later, and BANA was performed at the baseline and 8 weeks later for bacterial examination. During this period between treatment and measurement, they did not receive any special dental treatment and were instructed to take care of their oral cavity as usual. In this study, since the period of use of CHX was 1 week, there were no concerns about general side effects. However, participants who complained of side effects such as mouth irritation, tooth staining, dry mouth or decreased taste sensation during the study period were asked to immediately stop the study.

One examiner measured all indices during the study period, and the other was in charge of subject assignment, management, and education. They were blinded at each stage, and the examiner measuring the index was not aware of the group of subjects.

### 2.3. Clinical Measurements

Gingiva index (GI) was used for the assessment of gingival inflammation. Each of the four gingival areas of the tooth was given a score from 0 to 3 [8]. Dental plaque was assessed using the Turesky et al. Modified Quigley-Hein Plaque Index (TQHPI) after staining the tooth with a disclosing agent (2 TONE: Young, MO, USA), and was given a score of 0–5 by degree of staining [16]. To examine bacteria in the mouth, the microbial-enzymatic N-benzoyl-DL-arginine-2-napthylamide (BANA) test (BANA-Enzymatic test™ kit, Ora Tec Corporation, Manassas, VA, USA) was employed, which can detect causative organisms for periodontal diseases in 5 min. This tool’s working principle is a hydrolysis mechanism that causes a blue coloration with bacteria on the BANA strip [17]. The examiner assessed the change in color on a three-point scale: 0 for no change, 1 for pale blue and 2 for dark blue.

### 2.4. Statistical Analysis

The collected data were analyzed using SPSS Statistics 20.0 (SPSS Inc., Chicago, IL, USA) at a significance level of 95%. A normality test of the analyzed variables was performed using the Shapiro–Wilk test. To compare the changes of TQHPI and GI, repeated measured ANOVA was used at baseline, 4 weeks later, and 8 weeks later. Independent *t*-tests were performed to compare mean values of the two groups, and a paired *t*-test was conducted in order to compare before and after use of CHX products in the BANA test result.

## 3. Results

### 3.1. Changes of TQHPI and GI According to Treatment Period

The results of TQHPI and GI showed that there existed a significant difference in both the control group and test group depending on the period of treatments (*p* < 0.001). The mean TQHPI of the two groups were 1.00~1.04, but there was a marked decrease after 4 weeks, and a slight increase after 8 weeks. As a results of GI, the control group decreased after 4 weeks (1.23) compared to the baseline (1.77), and then increased again after 8 weeks (1.36). However, the test group appeared to be somewhat maintained even after 8 weeks (Table 1).

### 3.2. Comparison between Groups by ∆Value of before and after Treatment

The difference values for each period of TQHPI and GI were expressed as ∆value. There was no statistically significant difference in TQHPI between the control group and test group. However, the GI results showed that the test group had a higher ∆value than the control group, with a statistically significant difference in both ∆values after 4 weeks and 8 weeks (Table 2).

### 3.3. Bacterial Changes Pre -Intervention and Post-Intervention through BANA Test

As a result of the BANA test to evaluate changes in oral bacteria, the BANA score decreased to a statistically significant level after 8 weeks compared to the baseline in both groups (*p* < 0.001). After 8 weeks, the BANA score of the CHX gel group (0.44) was statistically significantly decreased compared to the CHX solution group (1.05) (*p* < 0.05) (Table 3).

## 4. Discussion

Although CHX is used as a representative antibacterial agent for periodontal treatment, it cannot be used for a long time in solution form due to side effects such as taste change and staining [18]. Recently, it has been developed in the form of a CHX gel or varnish, and it has been reported that there are no side effects and the antibacterial effect is superior to that of a CHX solution. CHX gel has been marketed in Korea since 2015, but clinical research is still lacking and its use in dental clinics is very low. Therefore, this study compared the clinical effects of CHX gel and solution on adults with gingivitis to increase the utilization of CHX gel in Korea.

The final 38 subjects were randomly divided into a CHX solution group and a CHX gel group, and the study was conducted for 8 weeks. In the previous RCT study [19] that evaluated the effect of CHX gel on peri-implant tissue, 40 subjects were recruited, and 37 subjects were finally selected.

This study included both clinical and microbiological evaluations in order to identify the efficacy of CHX products. GI and TQHPI were used for clinical evaluation in this study. The Gingival Index given by Löe and Silness make use of both visual assessment and bleeding on probing [20]. According to Cochrane Reviews, it has been reported that the GI has high-quality evidence in studies of gingivitis reduction in individuals with mild gingival inflammation [8]. In addition, TQHPI is a commonly used index to evaluate plaque removal in clinical studies [21].

It was found that there were significant changes in TQHPI and GI in the CHX gel and solution group according to the treatment period. Both groups showed a marked decrease after 4 weeks compared to the baseline, but slightly increased after 8 weeks.

Much research has demonstrated that the positive effects of CHX are considerable due to topical antiseptic action. Although the CHX solution also has staining problems, the research shows that CHX solution is the first product of choice when daily oral hygiene cannot be performed [22]. The CHX gel showed effects in several dental treatments at various concentrations such as 0.2, 1, and 2%.

According to Haraji et al. [14], a 0.2% CHX gel was reported to be effective in controlling for dry socket prevention after third molar surgery. On the other hand, 1% CHX gel has mainly been applied for peri-implant mucositis treatment and plaque formation prevention [23,24]. A study by Wang et al. [25] suggested that a 2% CHX gel is an effective root canal disinfectant. Although many studies have already demonstrated the efficacy of CHX gel, most have evaluated short-term effects. On the other hand, in this study, it was whether the effect continued for up to 8 weeks after using the CHX product for 1 week.

In this study, as a result of evaluating the GI ∆value after 4 and 8 weeks of the clinical trial, the CHX gel group showed statistically significantly higher effects than the CHX solution group, which indicates that CHX-gel has a long-term beneficial effect in reducing gingival inflammation. This is considered to result from an inhibitory effect on the formation of dental plaque while maintaining the treated area for a long time due to the viscosity of the CHX gel. The gel adsorbed onto dental tissues and mucous membranes and is gradually released during a prolonged period at the treatment area [26].

The CHX gel reduces gingival inflammation by killing bacteria. Because of its long-lasting effects it also prevents the build-up of plaque for up to 12 h after application [27]. Another study concluded, with a three-day non-brushing research design, that 1% CHX-gel application trays were more effective than 0.12% CHX dentifrice gel or regular dentifrice in inhibiting plaque accumulation [24]. Previous studies have suggested that CHX gel is an effective medication due to its broad antimicrobial spectrum [28,29]. Another study reported that application of chlorhexidine gel reduced inflammation and IL1-β levels in the peri-implant soft tissue [19]. Chlorhexidine’s antimicrobial mechanism is based on a process in which the cation compound becomes combined with enamel hydroxyapatite, acquired pellicle, bacteria & polysaccharide on dental plaque and, particularly, the oral mucous membrane. It helps to inhibit the build-up of bacterial colonies [30].

In this study, the BANA test was performed to identify the effect of reducing the causative bacteria of periodontal disease. The BANA test a modern method for bacterial culture. It detects the existence of three periodontal pathogens such as *P**. gingivalis*, *T**. denticola* and *T**. forsythia* in the subgingival plaque [31]. A previous study [32] reported that the use of the BANA Test as a chair-side test is recommended for proper diagnosis of periodontal disease and had good evaluation of treatment results. In this study, both the CHX solution group and the CHX gel group showed a statistically significant decrease in BANA Test score after intervention. In addition, the BANA score was statistically significantly decreased in the CHX gel group after 8 weeks compared to the CHX solution group. From these results, it was confirmed that the CHX gel was effective in reducing periodontal pathogens. These results are similar to the results of a previous study [33] confirming the effect of reducing alveolar osteitis of a CHX solution and CHX gel after third molar extraction. Other results showed that the 2% CHX gel produced the largest inhibition zones and was effective against all microorganisms [34]. Although the effect of a high concentration CHX gel was maximized, it was possible to prove the antimicrobial effect of the CHX gel.

This study confirmed the long-term clinical effects of 1% CHX gel, such as relieving gingivitis, inhibition of dental plaque formation, and reduction of bacteria that cause periodontal disease. In addition, there were no subjects who complained of side effects of the CHX gel during the study period, and since it showed several positive effects, the use of the CHX gel in dental clinics could be further expanded.

In recent years, the use of chlorhexidine has become more and more common in various medical fields. Chlorhexidine can also be of great help in the prevention of several systematic oral pathologies [35]. In particular, it is expected that CHX gel, which has a long-lasting effect, can be actively used in dental clinics.

### Limitations of the Study

This study has several limitations. First, it is difficult to generalize the study results due to the small number of subjects in a particular community. Second, it turned out that a relatively long period of the study resulted in the subjects’ lowered compliance and an unexpectedly high drop rate. Finally, this study may seem lacking in originality as similar studies have already been conducted. However, this study is meaningful in comparing the maintenance effect of CHX gel and solution. In future studies, it is necessary to confirm the short-term clinical effect using CHX gels with different concentrations such as 0.2, 0.5, 1, and 2%.

## 5. Conclusions

This study confirmed the clinical effect of CHX gel and solution. The 1% CHX gel has long-lasting antibacterial activity and has been shown to reduce plaque formation and gingival inflammation more than a 0.12% CHX solution. Therefore, the 1% CHX gel is expected to be actively used for non-surgical treatment of periodontal disease patients.

## Figures and Tables

**Figure 1 ijerph-19-09358-f001:**
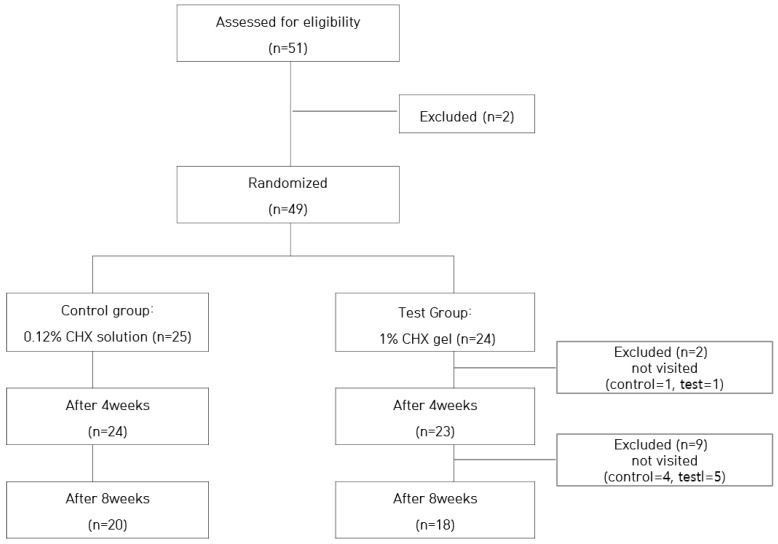
Flow chart of the participant selection process.

**Table 1 ijerph-19-09358-t001:** Change of PI and GI in CHX solution and CHX gel at the Baseline, 4 weeks and 8 weeks. Mean ± SD.

	Group	N	Baseline	4 Weeks	8 Weeks	*p*
TQHPI	Control	20	1.04 ± 0.29	0.59 ± 0.32	0.76 ± 0.22	<0.001
	Test	18	1.00 ± 0.29	0.46 ± 0.24	0.58 ± 0.26	
GI	Control	20	1.77 ± 0.25	1.23 ± 0.20	1.36 ± 0.23	<0.001
	Test	18	1.91 ± 0.15	1.21 ± 0.13	1.26 ± 0.27	

Control group: 0.12% CHX solution, Test group: 1% CHX gel; *p*-value was obtained by repeated measured ANOVA (*p* < 0.001).

**Table 2 ijerph-19-09358-t002:** Difference values of before and after treatment in PI and GI. Mean ± SD.

	Group	N	∆Value of 4 Weeks—Baseline	∆Value of 8 Weeks—Baseline
TQHPI	Control	20	0.44 ± 0.31	0.27 ± 0.36
	Test	18	0.53 ± 0.30	0.41 ± 0.37
	*p*		0.40	0.26
GI	Control	20	0.54 ± 0.22	0.41 ± 0.34
	Test	18	0.70 ± 0.15	0.65 ± 0.29
	*p*		0.01	0.03

Control group: 0.12% CHX solution, Test group: 1% CHX gel; *p*-value was obtained by independent *t*-test (*p* < 0.05).

**Table 3 ijerph-19-09358-t003:** Results of BANA test of CHX solution and CHX gel group. Mean ± SD.

Group	N	Baseline	8 Weeks	*p* *
Control	20	1.25 ± 0.71	1.05 ± 0.51	<0.001
Test	18	1.28 ± 0.82	0.44 ± 0.61	<0.001
*p ***		0.91	0.002	

Control group: 0.12% CHX solution, Test group: 1% CHX gel; * *p*-value was obtained by paired *t*-test (*p* < 0.001); *** p*-value was obtained by independent *t*-test (*p* < 0.05).

## Data Availability

The data presented in this study are available on reasonable request from the corresponding author.

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
