# Peer review of "Clinical Efficacy of 1% CHX Gluconate Gel and 0.12% CHX Solution: A Randomized Controlled Trial"

_ijerph, 2022, doi:10.3390/ijerph19159358_

Round 1

Reviewer 1 Report

Good day, 

Dear Authors,

Thank You for a pleasure to read this interesting work.

I have some comments to improve Your work.

Abstract 

Introduction. Please, add one sentence for actuality.

M&M

Group of people is not experimental it is control group.

Please, follow the structure for Abstract.

Introduction 

‘Periodontal disease is typical multifactorial disease: they can result from a variety of 25 causes, the main one being microorganism forming biofilm in the oral cavity’. Is it single or plural form?

Please, discuss here a possible dysbiosis of oral microflora after CHX use with references.

Materials and methods 

Sample size is considered like the amount of patients in EACH group but not for all groups especially You have RCT not case-control study. Please, also write the data You used from other research for sample size count.

To use ANOVA and t-test You need to have some requirements for distributions.

Please, add them in Statistics section.

Inclusion/exclusion criteria:

What comorbidities did You allow to be present in patients? and what did not?

Also, You need to mark primarily condition of oral cavity before CHX use.

Protocol

Did You recommend patients to use the same paste and the same diet?

Had You blinding on any stage?

Who measured the indices? What was intra- and inter-class agreement for results?

For comparison of results in dynamics for one group usually we use Wilcoxon-test. Why did You use other?

It is better for results also write correlations between indexes for both main groups (Pearson, for example).

Discussion

Please, explain why You chose exactly these indices of oral hygiene.

Please, write Your research limitations.

Conclusion

If You tell about Korean population, please, write appropriate part in inclusion

Criteria section.

Sincerely, Your Reviewer.

Author Response

Dear reviewer,

Thank you very much for your meticulous review. The revised version is attached, please check it.

Kind regards,

Reviewer 2 Report

1. Check spelling line 41 de-generation, line 44 treat-ment, these are just a few examples, there are a few more cases.
2. I would put table 2 all on the same page to make it easier to read.

Additional comments:

They can improve the explanation about the relevance of the study and why it is important to use gel vs. liquid solution, because in the discussion I don't see the advantage for the patient to use one system or the other, especially because the effect is really close between them. This topic is not new. There are a lot of different studies that show a better result from gel vs. mouthrinses.

Is probably that the statistics could be improved, or the explanation of them because they use scales to measure the index. First of all, they need to explain the test used to evaluate the normal distribution or not from the data, in each studied variable. After that, they need to explain better the statistics used, because they said that using ANOVA, but when we work with scales from 0 to 3, 4 or any other number really is not a value like in adhesion that each specimen has different results, we always work with the same values. In these cases the most common is used Kruskall-Wallis and DSCF to pairwise comparisons, and work with contingency tables. To the BANA test they said they used t-test, and we suppose that the different groups have a normal distribution, but this information is missing.

Author Response

(The authors gave the same response as above.)

Reviewer 3 Report

Dear Authors,

I would like to thank the authors for their efforts in connection with this randomized clinical trial. However, some questions need to be answered.

1. In the abstract, line 20, in my opinion the country (korea) should be omitted in the conclusions, unless they are comparable with other countries and also as an objective of the study.

2. In the last sentence of the first paragraph, line 35 and 36, the reference is missing, since it talks about the effectiveness of the use of chemical products.

3. In my opinion you should review the objective of the study, since what you have demonstrated is the efficacy of the gel compared to the solution, both of chlorhexidine.

4. The mean age was 46.3 years; however, the inclusion criteria refer to adults in their 20s. Normally at young ages they do not usually have probing pockets of less than 5 millimeters. He believes that age is important, depending on the type of periodontal disease. In the inclusion criteria he speaks of healthy patients, however, according to the probing pockets the patients had periodontal disease or were healthy. The risk factors within the inclusion and exclusion criteria are not clear.

5. Because it has not used an untreated control group. In this type of study it is very useful to compare whether the efficacy of the two treatments differs greatly from the normal hygiene established without treatment.

6. The differences shown in Table 2 are clinically noticeable in the patients or are only statistical results.

7. The application of clohexidine gel and solution was the first week. However, the measurements were at 4 weeks and 8 weeks. What hygiene standards were the patients told to have during this period.

8. The sample was considerably reduced from the 4-week measurement to 8 weeks. He believes that this may influence the statistical variables, which he considered to be a bias in the study.

9. The conclusions reached should be evaluated since the objective of the study was to compare the two treatments, not only the chlorhexidine gel.

Best Regards

Author Response

(The authors gave the same response as above.)

Reviewer 4 Report

Dear authors,

The article entitled Clinical Efficacy of 1% CHX gluconate gel: a Randomized Controlled Trial brings some useful information for the clinical practice, to improve the results of anti-bacterial activity.

Overall, the study is limited and does not contain much data, so it is necessary to supplement it with more information.

Here are some recommendations for the improvement of the paper:

-the paper needs English correction

-the Introduction section should include more data upon CHX gel - such as, available concentrations etc.

-lines 29 and 38: please consider "plaque" instead of "plague"

-the null hypothesis and the test hypothesis should be specified

-the Materials and method section - subsection 2.2: Clinical Protocol - the amount of gel used should be specified, and if the gel was applied only to the dental and gingival surfaces or to the entire surface of the oral mucosa, if it was applied only where there were periodontal pockets or everywhere

-the placement of numerical values at the beginning of the sentence should be avoided - eg, line 69

-the Results section - lines 119-120 - please rephrase, as the expression is unclear

-the expression of the same data in both tables and graphs should be avoided. Please choose one of these data rendering modes

-please include the "mean and SD" specification in all tables

-the Discussion section: lines 167-169 - please rephrase, as the expression is unclear: is it an increase, or a decrease?

-this section should not contain a repetition of the data presented in the Results section, but it should contain more comments on the obtained results and comparisons with similar studies in the literature

-please consider that the effect of preventing the build-up of plaque for up to 12 hours after application is characteristic to CHX regardless of the form of presentation. At the same time, when comparing the results, it should also be borne in mind that the concentration of CHX in the gel is higher than in the solution.

-the Conclusion section: the results of the study are not sufficient to be able to make recommendations regarding the use of the gel in a certain geographical region.

Author Response

(The authors gave the same response as above.)

Round 2

Reviewer 1 Report

Dear Authors, 

Thank You for Your great work for manuscript correction.

Although, there are yet several points that must be changed.

Abstract

M&Ms. We treat patients not the ‘subjects’.

M&Ms

‘In the previous RCT study [16] that evaluated the effect of CHX 75 gel on peri-implant tissue, 40 subjects were recruited and 37 subjects were finally selected. 76 Fifty-one participants were screened for this study, and 2 people were dropped out ac- 77 cording to the inclusion and exclusion criteria. The selected 49 subjects were randomly 78 divided into two groups (control group:25, test group: 24). During the study period, 11 79 patients were excluded for reasons of giving up or not visiting the clinic center’. This fragment is more appropriate for discussion. For sample size we use not amount of patients from previous study but criterion that is exactly primarily finite point (for example, mean and SD of appropriate bacteria amount). We use statistics program, online-calculator or count by hand with the traditional formula for sample size count.

Please, re-write. 

Inclusion criteria 

‘Inclusion Criteria: good systemic and oral health..’ It is not scientific determination.

For oral hygiene it can be good but for general condition and comorbidity we use other words. Please, re-write.

Also, if we tell about ‘good health’ for oral cavity then the use CHX has no the indication.

Please, write ‘Study limitations’ as separated point not inside of Discussion.

Sincerely, Review

Author Response

Dear reviewer,

Thank you very much for your second review. The revised version is attached, please check it.

Kind regards,

Reviewer 3 Report

Dear Authors,

The article has been improved, and the questions I asked have been answered. In my opinion it may be suitable for publication in the journal.

Best Regards

Author Response

Dear reviewer,

Thank you very much for reviewing our paper

Kind regards,

Reviewer 4 Report

Dear Authors,

Thank you for taking into account most (but not all) of the recommendations I made in the previous review of your article.

Unfortunately, the first and most important observation of my previous report, namely the one regarding the fact that the study is limited and does not contain much data, remains valid. I still consider that it is necessary to supplement it with more information, among which I could suggest completing the research with aspects regarding the possible side effects felt by the patients during the study; or the inclusion of the results of this research in a review-type article, in which all the positive and negative aspects of the use of these products can be presented, especially since the stated purpose of the article is to lead to the expansion of the use of CHX gel in dental practice.

Author Response

(The authors gave the same response as above.)
